# Irisin in Reproduction: Its Roles and Therapeutic Potential in Male and Female Fertility Disorders

**DOI:** 10.3390/biom14101222

**Published:** 2024-09-27

**Authors:** Muhammad Ibrahim Khan, Muhammad Imran Khan, Fazal Wahab

**Affiliations:** Department of Biomedical Sciences, Pak-Austria Fachhochschule: Institute of Applied Sciences and Technology, Mang, Haripur 224000, Khyber Pakhtunkhwa, Pakistan; b20f0217bms029@fbse.paf-iast.edu.pk (M.I.K.);

**Keywords:** irisin, reproduction, spermatogenesis, polycystic ovary syndrome, fertility, reproduction, HPG axis, gestational diabetes mellitus, IUGR

## Abstract

The current study focused on identifying the potential of irisin in mammalian reproduction. The established role of irisin, a proteolytic product of FNDC5, in adipose tissue browning, energy metabolism, and thermogenesis suggests its role in reproductive health, often disturbed by metabolic imbalances. Various studies on mice demonstrated irisin′s role in improving spermatogenesis, sperm count, and testosterone levels by influencing the hypothalamus–pituitary–gonadal axis. Moreover, in females, there is a fluctuation in levels of irisin during critical reproductive stages, including menstrual cycles, puberty, and pregnancy. Conditions like pregnancy complications, precocious puberty, and polycystic ovary syndrome (PCOS) are found to have an association with abnormal irisin levels. The potential role of irisin in endometrial receptivity and preventing endometritis is also discussed in this review. Overall, the influence of irisin on female and male reproduction is evident from various studies. However, further research is needed to elucidate irisin mechanism in reproduction and its potential as a therapeutic or diagnostic tool for reproductive dysfunctions and infertility.

## 1. Introduction

Reproduction in mammals is sexual; females and males equally contribute to genetic information [1]. The coordination of the vascular, neurological, and endocrine systems contributes to the complex process of sexual health [2,3]. Fertility and reproduction are linked to the endocrine function and energy metabolism of adipose tissue [4,5]. Adipose tissue releases various chemokines, hormones, proteins, cytokines, growth factors, and complementary factors involved in angiogenesis, vascular hemostasis, reproduction, and lipid metabolism [6]. Endocrine functions of adipose tissue are exerted by the systemic and local biological functions of these factors [7]. Irisin (proteolytic protein) is a myokine induced by exercise [8], stored in muscles as a precursor form, and part of the Fibronectin Type III Domain containing 5 (FNDC5) protein [9,10]. It is produced by the cleavage of the FNDC5 protein [4,10]. Irisin is secreted from visceral adipose tissue, skeletal muscle, brain, subcutaneous tissue, testis, liver, and other organs. According to new studies, endometrium and ovaries are also capable of producing irisin [4]. It is involved in energy homeostasis, thermogenesis, and adipose tissue browning [11]. Irisin has a role in energy homeostasis and male fertility [12]. It can reverse male infertility caused by metabolic syndromes and enhance steroidogenesis and spermatogenesis via the direct or indirect beneficial impact of modifying inflammation, insulin resistance, testicular functions, and an imbalanced HPG axis [12,13].

Insulin resistance is one of the complications associated with obesity [14]. Hyperinsulinemia linked with insulin resistance leads to elevated levels of serum testosterone and luteinizing hormone, resulting in infertility in females [15]. Studies have shown the role of irisin in significantly lowering the elevated luteinizing hormone, BMI, and serum insulin levels in obese female mice [16].

Studies on mice showed the relationship between disordered endocrine metabolism and irisin deficiency, resulting in poor development, growth, and decreased fertility. Irisin promotes the secretion of integrin αvβ3 and LIF, thus improving the implantation of blastocysts in rats with implantation failure. Therefore, the endometrium’s poor receptive state is improved [17]. The differentiation and improvement in the function of trophoblast in the human placenta are promoted by introducing recombinant irisin in abnormal placentation [18]. Irisin level in normal pregnant women is higher than in non-pregnant women [19,20]. Women with preeclampsia had lower irisin levels than normal pregnant women [19,21] and this difference becomes most significant during the third trimester [19]. Moreover, serum irisin level is also increased in polycystic ovary syndrome patients, which may be attributed to insulin resistance [22]. However, a significant reduction in irisin level was observed in patients who had been on treatment with metformin [23,24].

Serum irisin level is also associated with the growth regulation of fetuses in early infancy and in utero [25]. According to studies, increased maternal serum irisin concentration and lower fetuin-A levels during the first trimester of pregnancy are correlated with the birth of macrosomic infants [26]. Studies also have found a significant relationship between the genetic variation in maternal FNDC5 rs726344 and preterm birth [27].

## 2. Structure of Irisin

X-ray crystallography and biochemical studies on irisin revealed the function and structure of irisin [28,29]. According to that, the folding of irisin occurs similarly to the folding of fibronectin III (FNIII) domain-containing proteins. However, demA tight dimer of eight-stranded β-sheets is formed by the association of two subunits as shown in Figure 1. The interactions between continuous β-sheets give rise to ten hydrogen bonds between subunits in the dimer, which is responsible for its thermostability. Moreover, other interactions like the ‘tryptophan zipper-like interaction’ and ‘salt bridge formation’ between the side chains of two subunits contribute further stability [29,30]. Moreover, experiments suggest the non-glycosylated and glycosylated forms are dimers. The dimer form of irisin 59 is also supported by mutagenesis and structural data. Moreover, the candidates for interaction with unidentified putative receptors are the N-terminus, loop 106–108, and region 55–58 of irisin. In its preformed dimeric form, irisin binds to its putative receptor [29].

## 3. Physiological Functions of Irisin in Various Organs

Different studies have confirmed the multifunctional and beneficial role of irisin (Table 1) in body homeostasis [31]. It plays a crucial role in the body by its effect on different organs and tissues such as brain, bone, liver, pancreas, or fat tissues, as shown in Figure 2, through autocrine, paracrine, or endocrine signaling pathways [32]. Irisin is a myokine regulated by exercise and provides beneficial effects of exercise such as thermoregulation and weight loss in humans [10,33].

Irisin promotes the conversion of white adipose tissue into brown adipose tissue by browning white adipose tissue. This phenomenon leads to an increased metabolism of glucose and lipids and increased energy expenditure [41]. The conversion of WAT to BAT is promoted by irisin in mature fat cells by increasing uncoupling protein (UCP-1) expression. The MAPK phosphorylation induced by irisin regulates browning [42,43]. Moreover, the reduced formation of new adipocytes and inhibition of adipogenesis in WAT preadipocytes is also promoted by irisin [43]. Irisin also promotes synaptic plasticity and neurogenesis in the hippocampus by increasing the expression of BDNF (brain-derived neurotrophic factor). BDNF is a signaling molecule for neurogenesis and synaptic plasticity [44,45]. Therefore, irisin has a beneficial role in protection against neurodegenerative diseases and improvement in cognitive function [31,38,46,47]. Irisin enhances neuronal migration and survival and reduces neuronal damage induced by oxidative stress [31,48]. Moreover, irisin provides an antidepressant effect by activating the PGC-1α/BDNF pathway [44,49]. Irisin reduces the risk of bone fractures and bone loss associated with aging and stimulates bone formation and mineralization, therefore contributing to bone health [39,50,51]. It contributes to expressing bone-specific genes and activation of osteoblasts [52]. It promotes differentiation of osteoblasts [53] and increases the aerobic glycolysis pathway within the osteoblasts [54]. Osteogenesis is promoted by the irisin involving MAPK signaling pathway [53]. In addition, irisin inhibits the secretion of pro-inflammatory cytokines like IL-6, IL-1β, and TNF-α and reduces systemic inflammation. Irisin reduces the oxidative stress in macrophages and increases antioxidative enzyme expression, thereby exhibiting antioxidative property [31,37,55]. Moreover, the dysregulated levels of irisin in obesity is observed. Adipocyte function modulation, improving insulin sensitivity, and decreased inflammation in adipose tissue can be promoted by irisin. The two adipokines: adiponectin and leptin that are associated with metabolic syndrome and obesity are also expressed by the effect of irisin. However, the exact role and mechanism of irisin in obesity is still unknown [31]. Due to its anti-inflammatory properties, its role in energy homeostasis, browning of white adipose tissue, and metabolic improvement, irisin helps in maintaining cardiovascular health and prevent arterial stiffness and cardiovascular diseases [40]. Some studies have also suggested that in addition to adipose tissue browning, improving metabolism and insulin sensitivity, irisin may provide protection against certain types of cancers by inhibiting the secretion of proinflammatory cytokines, inducing apoptosis, and inhibiting tumor growth [34,35,36,56]. However, there are conflicting findings as well in this regard suggesting its role in promoting metastasis and tumor growth in certain contexts. Further studies are needed to find the exact role of irisin in carcinogenesis. Irisin can be used as a potential therapeutic target for conditions such as bone disorders, neurodegenerative diseases, diabetes, and obesity [31].

## 4. Irisin Role in Various Aspects of Reproduction

### 4.1. Hypothalamic–Pituitary–Gonadal Axis

The hypothalamus–pituitary–gonadal axis (HPG) is the control unit that involves molecular signaling from hypothalamus to pituitary and towards the gonads, thus critically important in regulating reproductive function [57].

Various studies have reported potential involvement of irisin in HPG axis activity (Figure 3). There are three defined activity periods exhibited by the HPG axis. One during fetal life, then in minipuberty during postnatal life, and the third one during the onset of puberty. HPG axis activity stops at the end of postnatal development during the juvenile phase and is reactivated at the end of the juvenile phase during the onset of puberty [58]. This mechanism of reactivation of HPG axis activity during pubertal stage is unclear [59]. According to one hypothesis, there is a certain crucial level of body muscle or body fat that is essential for the onset of puberty and supporting reproduction. Puberty onset is triggered by the hypothalamic neuron network receiving the metabolic signal when body fat and body muscle reach a critical level that is necessary for puberty onset [60]. Irisin can have a role in activation of the HPG axis during pubertal onset [61,62]. There is a variation in plasma levels of irisin reported with different stages of puberty [63]. The involvement of irisin in HPG axis activation during puberty is further suggested by the study in which both female and male monkey hypothalami were found to have increased abundance of FNDC5 transcripts around the onset of puberty [61]. The ARC and ventromedial nucleus of the hypothalamus in rhesus and marmoset monkeys are found to have noted irisin expression. These regions are responsible for regulation of feeding, energy homeostasis, and reproduction [64]. There is a putative interaction between irisin neurons and GnRH neurons in the hypothalamus of the rhesus monkey. In the hypothalamic neurons of mice, the GnRH release is regulated by the involvement of irisin. This suggests the potential role of irisin in the onset of puberty and reproductive physiology [61].

An in vitro study using murine pituitary cells demonstrated that GnRH treatment increased LH production in a dose-dependent manner, while irisin alone significantly increased LH production from murine pituitary cells. However, combined treatment with GnRH and irisin did not produce this effect [65].

A study on intracerebroventricular administration of irisin and its effect on HPG axis has shown that irisin infusion reduced the hypothalamic GnRH mRNA and its protein expression. Moreover, it also caused a lowering of serum FSH, LH, and testosterone levels in male rats [66]. Conversely, other studies showed that irisin promotes secretion of LH and FSH in female rats [67]. In fact, irisin absence in female mice was found to have lowered levels of LH and FSH as compared to wild-type mice [4]. These findings suggest that FSH and LH expression could be promoted by irisin, whereas it could also inhibit LH and FSH secretion via competing with GnRH. This dual activity could explain the fluctuating hormone levels, depending on the dominant effect at a given time [62].

### 4.2. Irisin Role in Male Reproduction

The data suggesting a linkage between male reproductive function and irisin is scarce. However, its secretion from the penis, seminal vesicle, and testis suggests its paracrine and autocrine effects on the functioning of male reproduction [12]. Various studies have found an association between irisin function in energy homeostasis, inflammation, obesity, and male reproduction, as shown in Figure 4. Male reproductive function is regulated by an endocrine regulator, the HPG axis [12]. The hypothalamus sends the signal to the pituitary gland via GnRH [68]. From the pituitary, the signal is sent to the testis by the release of FSH and LH. The effect of LH on the testis is exerted by its binding to LH-R, expressed in interstitial Leydig cells. On the other hand, the FSH effect on the testis is exerted by its binding to FSH-R, expressed within seminiferous tubules in Sertoli cells [69]. The conversion of testosterone from cholesterol takes place in response to LH signaling through a series of biochemical reactions [70]. There is an association between energy dyshomeostasis and upregulation of the expression of estrogen receptors within the male hypothalamus [71]. In turn, the negative feedback mechanism triggers and inhibits the release of GnRH, leading to a decline in the release of LH and FSH, resulting in impaired production of testosterone. The aromatase level increases from energy dyshomeostasis, promoting estrogen conversion from testosterone, resulting in androgen suppression and inhibition of testicular function [72]. Irisin, via the elevation of mitochondrial UCP1 expression, activates lipolysis and thermogenesis, resulting in energy balance maintenance. This might cause the downregulation of estrogen receptor expression in the hypothalamus and ensure optimal GnRH pulsative release and subsequent FSH and LH release, leading to normal testosterone production. Irisin also re-establishes the energy balance and represses aromatase activity, thus inhibiting estrogen conversion from testosterone [12,43,73]. Spermatogenesis is adversely affected due to energy dyshomeostasis via testosterone suppression [74]. In obese male rats’ administration of irisin causes downregulation of IR and decreased BMI and increases serum levels of LH and FSH and increased testosterone level, causing improvement in spermatogenesis as well as increased sperm parameters like sperm motility and count [75]. Another invitro study suggested irisin role in spermatogenesis because of increased irisin expression in undifferentiated spermatogonia transcript and Sertoli cells in organotypic testicular culture in primates. In rhesus monkeys, irisin and FNDC5 mRNA levels vary between non-breeding and breeding seasons, suggesting a link to spermatogenic activity [76]. In organotypic cultures, irisin treatment upregulated genes associated with spermatogonial activity (SALL4, cKIT, ID4, and KLF4) and Sertoli cell function (GDNF). These findings suggest that expression of irisin appears to be dynamically regulated and potentially influencing spermatogenesis [76].

Obesity is one of the common causes of male infertility [77]. Testicular apoptosis and oxidative stress play a significant role in obesity-induced spermatogenesis dysfunction [78]. Oxidative stress is one of the pathological factors responsible for male infertility [79]. For acrosomal reactions, hyperactivation, and capacitation of sperm, lower levels of ROS are necessary. Higher levels of ROS can disturb the reproductive function of obese males through DNA fragmentation of sperm, apoptosis of testicular cells, and lipid peroxidation [80]. Irisin is found to reduce the ROS levels and increase the expression of Nrf2 and its respective downstream antioxidants, therefore providing protection against testicular damage induced by obesity [81]. Moreover, oxidative stress can also induce endoplasmic reticulum (ER) stress, which is involved in particulate matter 2.5-induced reproductive toxicology [82]. The ER stress markers are found to be elevated in obese mice that are fed a high-fat diet. However, irisin treatment decreases the level of ER in obese male mice [83]. Leydig cells in the testis are responsible for the production of testosterone, which is crucial for spermatogenesis. Obesity reduces testosterone levels by impairing Leydig cells’ function [84]. In obese mice, irisin treatment increases steroidogenic enzyme expression in the testis. Additionally, irisin can protect Leydig cells against endoplasmic reticulum (ER) stress and oxidative damage induced by palmitic acid (PA), a saturated fatty acid [81,85].

A study on human subjects has shown that serum testosterone levels, sperm motility, and count are decreased in obese subfertile males compared to normal lean. Moreover, in obese patients, plasma irisin levels were significantly decreased. In addition, a negative correlation was found between plasma irisin levels, sperm count, and motility [81]. Furthermore, a mouse study involving feeding a high-fat diet developed an obesity-induced spermatogenesis dysfunction model [86]. The body weight, plasma insulin, and cholesterol levels were elevated in mice that were fed a high-fat diet. However, the irisin treatment decreased the body weight and plasma insulin and cholesterol levels in high-fat diet-fed mice [81]. Additionally, FNDC5 protein expression in testis and serum irisin levels were lowered in obese mice. Moreover, HFD promoted fewer sperm in atrophied seminiferous tubules and thinning of the blood–testis barrier in obese mice. However, irisin treatment attenuated these pathological changes that were induced by HFD [87]. Irisin increased the seminiferous tubule diameter and promoted blood–testis barrier improvement. Moreover, serum testosterone levels were significantly decreased, and sperm viability, motility, and count were decreased in obese mice that were fed a HFD [81,87]. However, irisin treatment improved the serum testosterone levels and improved sperm viability, motility, and count [75]. The protective role of irisin against obesity-induced spermatogenesis is mainly through activation of the AMPKα pathway [88]. However, in AMPKα deleted mouse models, the protective effects of irisin against spermatogenesis dysfunction and testicular damage were abrogated, suggesting the dependence of irisin on AMPKα to promote such protective effects [81].

Another study on obese male mice has shown that irisin improves measures including sexual performance, ejaculation, and penile erection and decreases testicular damage induced by fat. Therefore, potency, libido, and sexual performance that are affected by a high-fat diet were restored by irisin [87]. Selective serotonin reuptake inhibitors (SSRIs) like paroxetine are found to cause various types of sexual dysfunction in males, including delayed ejaculation, impotence, anorgasmia, reduced sexual satisfaction, and reduced sexual desire [89]. According to a study, circulating irisin levels are lowered and sexual behavior is impaired in paroxetine-treated male rats. However, administration of irisin in these rats reversed sexual dysfunction and improved motivation and copulatory performance. Moreover, paroxetine reduced the expression of key genes in brain regions (medial preoptic area and nucleus accumbens) that are linked to sexual function. Irisin treatment increased the expression of genes in these brain regions and increased the decreased level of testosterone in paroxetine-treated mice [90].

In another study on Egyptian men with metabolic syndrome, serum testosterone and irisin levels were measured and corelated. According to this study, a positive correlation was found between circulating irisin levels and serum testosterone levels, which were significantly lowered in patients with metabolic syndrome [91]. Moreover, another study investigating the effect of testosterone treatment on irisin levels in men with late-onset hypogonadism and metabolic syndrome has also shown a positive correlation between testosterone and circulating irisin levels, both of which are found to raise on testosterone replacement therapy [92].

All these studies show the promising role of irisin in improving male reproductive function. Irisin role in energy homeostasis, reducing oxidative stress, improving spermatogenesis, increasing testosterone levels, sperm count, and motility suggests its significant contribution in maintaining male reproductive health and its possible therapeutic potential for treating male infertility. Further studies involving clinical trials are needed to confirm findings from animal studies and further understanding of underlying mechanisms. Further research is needed to find its interaction with other hormones and signaling pathways in the male reproductive system.

### 4.3. Irisin Role in Female Reproduction

In females, there is variation in secretion of irisin across different phases of life from puberty to the menopause stage or through different stages like near childbirth or pregnancy, suggesting its effect on female reproduction [62]. Moreover, irisin circulation changes in cyclic pattern during the menstrual cycle. During the luteal phase, the circulating irisin increased by 25% more than that in the follicular phase, but the irisin level gradually decreased and returned to baseline during the early follicular phase with the regression of the corpus luteum [19,62]. The increased irisin level during the luteal phase suggests its role in the ovarian cycle [19]. Moreover, integrins are the receptors for irisin, and steroidogenesis can be altered by it in the granulosa cells of the ovaries [93]. Another in vitro study involved the effect of irisin, insulin, or a combination of both on human ovarian granulosa cells, and estrogen production was measured. Results showed that irisin treatment on human granulosa cells increased estradiol production by 2.5 folds, while insulin treatment alone increased the production by 1.4 folds. However, the combined treatment with insulin and irisin abolished the stimulatory effect in estradiol production [65]. Moreover, irisin also improves placental and uterine function in women [93].

The significant increase in irisin level is observed in pubertal and pregnant women with the activation of the HPO axis. The mechanism of irisin increase with sex hormone level is not clear [62]. Irisin deletion leads to decreased levels of follicle-stimulating hormone, estradiol, and luteinizing hormone, while increasing progesterone levels [4,93,94]. Moreover, irisin deletion also caused impaired development, growth, and fertility in mice while increasing the mortality rate. Ovarian development, ovulation, and luteinization were also deleteriously affected because of irisin deletion. A significant decrease in the ratios of antral or secondary follicles to total follicles was observed in female mice that underwent irisin deletion [4]. Moreover, irisin deletion also caused impairment of the estrous cycle [4,93].

As obesity increases serum testosterone, insulin, and luteinizing hormone levels, the introduction of recombinant irisin significantly lowered the serum luteinizing and insulin levels in female obese mice. Irisin also lowered the testosterone level in obese female mice, but not to a significant extent. Therefore, irisin may reverse obesity-associated infertility in females [16].

Another study on females with central precocious puberty has shown a higher level of irisin in girls with central precocious puberty than the control group. A positive correlation was also found between serum irisin level and puberty markers like height-SDS, ovary size, weight-SDS, body mass index standard deviation scores (BMI-SDS), baseline FSH and LH, and bone age. The puberty markers along with irisin were found to be raised in central precocious puberty. This suggests a potential role of irisin in reactivation of the HPG axis and onset of puberty [95].

These findings, drawn from both historical and recent studies, highlight the multifaceted role of irisin in regulating and supporting female reproductive functions across various stages of life and physiological conditions (Table 2 and Figure 5).

### 4.4. Irisin in Pregnancy

Serum irisin level was found to be higher in pregnant women than in non-pregnant women [96]. The Irisin level during the mid-pregnancy period is 16% higher than the early pregnancy period, and this level increases to 21% during late pregnancy [19]. During pregnancy, the irisin level shows a positive correlation with HOMA IR (Homeostatic Model Assessment of Insulin Resistance) [19]. The increased insulin resistance in normal pregnant women may be attributed to increased irisin levels. This suggests that irisin may be involved in metabolic changes during normal pregnancy, including an insulin-resistant state during pregnancy [97]. Specific FNDC5 immunostaining is shown by the human placenta, which may be responsible for increased serum irisin level during pregnancy [19]. The increased irisin level during pregnancy may be a compensatory response to irisin resistance [98], and hence decreased fat browning and decreased thermogenesis may be related to the lower temperature with the progression of gestation [19].

#### 4.4.1. Irisin Concentration in Predicting Macrosomia, Fetal Birth Weight, and Growth Pattern

Macrosomia is an obstetric life-threatening condition for both the fetus and the mother. It refers to excessive growth of the fetus, which usually has a high birth weight irrespective of gestational age [99]. According to studies, first-trimester serum irisin concentration can help in predicting macrosomia [26,100]. A positive correlation of maternal serum irisin concentration with the birth weight of the fetus was found [100]. The serum concentration of irisin is significantly higher in the mothers of macrocosmic infants than in the mothers of normal infants during the first trimester of pregnancy [26]. Moreover, PIGF was also raised in maternal serum. However, the most prominent marker for predicting macrosomia is found to be irisin [26]. It is suggested that increased irisin resistance due to compensatory mechanisms may be the reason for the deprivation of the beneficial effects of irisin [26].

Al-Maini EH studies have revealed the correlation between maternal or fetal irisin concentration with the large, adequate, and small for gestational age fetuses. They have found that the irisin concentration was significantly lowered in fetal blood with intrauterine growth restriction than in normal fetuses [25]. Moreover, the irisin concentration in maternal blood was also significantly lowered in IUGR fetuses than in mothers with normal fetuses [101]. The maternal blood irisin concentration gives an excellent prediction for intrauterine growth restriction. Therefore, irisin concentration in maternal or fetal blood is directly correlated with fetal birth weight and growth pattern [102]. Higher levels of serum irisin are associated with macrosomia, whereas lower concentrations are linked with IUGR. Therefore, irisin could serve as a biomarker for assessing fetal health and growth throughout pregnancy (Table 3).

#### 4.4.2. Irisin and Endometrium

During implantation, the embryo gets attached to the endometrium of the uterus [103]. For successful implantation, two prerequisites are essential: a receptive endometrium and an energetic blastocyst [103,104]. Recent studies have expanded our understanding of irisin′s role in endometrial function, particularly highlighting its effects on embryo implantation and inflammation regulation [24]. The receptivity of endometrium can be determined by some markers like LIF, integrin αvβ3, ovarian hormone, and so on [105]. Mifepristone acts as an antagonist for progesterone, resists implantation of the blastocyst, and destroys endometrium receptivity. It also lowers the level of endometrial integrin αvβ3 and LIF [106]. A study on metformin-treated and irisin-treated mice groups has revealed that irisin increased the implantation receptivity of endometrium and therefore increased the number of blastocysts implantation in the irisin group than in the Mifepristone group. The reduced integrin αvβ3 and LIF due to Mifepristone can also be improved by irisin (Figure 6) [17].

Another study on female mice suggests that endometritis caused by lipopolysaccharide (LPS) can also be reversed by irisin [107]. The study showed that LPS resulted in obvious uterine damage, and [107], the levels of NF-κB and inflammatory mediators (iNOS, COX-2, IL-1β, TNF-α, and IL-6,) were significantly increased in uterine tissue [108]. However, there was a reduction in adenosine monophosphate-activated protein kinase (AMPK) levels. The irisin pretreatment reversed the LPS-induced phenomenon. The effect of irisin on the uterus was blocked by compound C (AMPK inhibitor) [107]. Therefore, AMPK-NF-κB pathway regulation underlies the beneficial functions of irisin, as shown in Figure 6. Irisin pretreatment also reversed the oxidative factor alterations that were caused by LPS, therefore suggesting the antioxidation role of irisin along with the anti-inflammatory role [107].

Irisin has been found to enhance the receptiveness of the endometrium to implantation by increasing levels of crucial proteins such as LIF and integrin αvβ3. These proteins are markers of endometrial receptivity, which is crucial for successful embryo implantation. Research in rats has shown that irisin treatment can counteract the negative effects of Mifepristone, a drug that typically decreases the levels of these proteins and thus impedes implantation [17]. Additionally, irisin has shown promising anti-inflammatory properties in various tissues, including endometrium. It can reduce the progression of inflammation by modulating cytokine production, which could potentially aid in managing endometrial inflammation [109].

Overall, irisin influence on the endometrium appears to be multifaceted, contributing both to enhancing the environment for embryo implantation and managing inflammation (Appendix A), which could have implications for treating conditions like endometritis and other inflammatory disorders of the reproductive system.

### 4.5. Irisin in Gestational Disorders

#### 4.5.1. Preterm Birth

Preterm birth is the birth of a newborn before completion of 37 weeks of gestation and is associated with infant morbidity and mortality [110]. The fetal or maternal FNDC5 gene is a precursor of irisin. Its polymorphism is found to have an association with preterm birth [27]. The two different alleles of FNDC5, namely, rs726344 G and rs1746661 A, of neonates and mothers were studied by RFLP. The women with the genotype of FNDC5 rs726344 GG were found to have increased chances of delivering preterm neonates by 2.18 folds when compared to AG or AA genotypes. Moreover, the neonates with genotype FNDC5 rs726344 G had 2.28 folds increased chances of being born preterm, as shown in Figure 7. Therefore, FNDC5 rs726344 G polymorphism and increased risk of preterm birth have a significant association (Appendix A) [27].

#### 4.5.2. Irisin and Preeclampsia

Preeclampsia is a pregnancy-specific disorder characterized by hypertension and proteinuria after 20 weeks of gestation in women who were previously normotensive [111]. It has an impact on perinatal and maternal mortality and morbidity [111]. Studies have shown the irisin level during the third trimester to be lowered in patients with preeclampsia. However, studies on irisin levels in severe and mild preeclamptic patients were not significant [21]. Patients who had cesarian c-section had lower serum irisin levels compared to those with vaginal deliveries [112]. Thus, irisin levels can be used as preeclampsia biomarkers [21,112]. All these studies suggest a potential association of irisin with preeclampsia (Figure 8). The exact mechanism of irisin role in preeclampsia is not yet fully understood and therefore needs further research.

#### 4.5.3. Irisin and Gestational Diabetes Mellitus

Irisin is associated with improved glucose metabolism and insulin sensitivity, playing a role in the management and prevention of metabolic disorders like diabetes [113]. Gestational diabetes mellitus is the diabetes that develops in pregnant women during pregnancy [114,115]. It is characterized by high blood glucose levels that can be a risk for both the mother and developing baby [115]. The exact cause of gestational diabetes mellitus is not understood. During pregnancy, the changes in hormonal levels, such as increased levels of hormones, which are insulin-blocking, can play a role in the development of GDM [116]. Studies have shown a significant decrease in irisin levels in the blood of women with gestational diabetes mellitus as compared to normal pregnant women. However, in postpartum, no significant difference in irisin levels was observed between the two groups [117]. This suggests the beneficial role of irisin in protection against the development of gestational diabetes mellitus [117,118]. Further studies have shown that decreased irisin level during the first trimester of pregnancy is helpful in an independent prediction of gestational diabetes mellitus [119], therefore helpful in taking early prevention strategies. Further research is required to fully understand the mechanism of irisin in preventing gestational diabetes mellitus, its potential as a therapeutic agent and early prediction biomarker (Figure 9 and Appendix A) [117,118,120]. Recent studies underscore the complexity of GDM and the potential of irisin and other biomarkers in managing and predicting this condition (Figure 9). For instance, exercise has been highlighted as a beneficial factor in maintaining healthy irisin levels, which could be protective against GDM [98]. Additionally, emerging research points to various lesser-known biomolecules that could play a role in the pathogenesis of GDM, indicating new directions for future research and potential early diagnostic strategies [121].

### 4.6. Irisin and Polycystic Ovarian Syndrome

Polycystic ovarian syndrome (PCOS) is a complex heterogeneous endocrine disorder in females characterized by clinical manifestations of polycystic ovaries, anovulation, and hyperandrogenism [122]. For PCOS occurrence, metabolic disorder is the key factor [123]. Recent studies have suggested that irisin may play a role in the development of polycystic ovary syndrome, as a significant association has been found between PCOS and irisin concentration in PCOS patients (Appendix A) [124,125,126]. A significant increase in irisin concentration was observed in PCOS patients than in the control group [24]. Moreover, a positive correlation was noted between irisin concentration and the severity of insulin resistance [24]. This suggests the involvement of irisin in the pathophysiology of PCOS [124,127,128].

BAT has a role in metabolic balance via heat production [129]. The involvement of BAT in the metabolic disorder of PCOS is also demonstrated, where a significant decrease in activity of BAT was observed in PCOS patients [130]. Moreover, a recent study on PCOS mice has proved that irisin activates BAT function, thus improving insulin sensitivity and thermogenesis. Moreover, irisin treatment reduces ovarian cystic follicle formation and improves disorders of steroid hormone disorder and menstrual cycle [127]. Another study in Baghdad reported that four months of treatment of PCOS patients with metformin significantly reduced the increased irisin level. Insulin resistance can be predicted in PCOS patients by a simple irisin test [23].

These results suggest the significant role of irisin in the pathophysiology of PCOS and the possibilities of irisin being used as a therapeutic tool for the treatment of PCOS. Further research is needed to find the exact role of irisin in PCOS and to develop effective treatments. Irisin plays a role in reducing insulin resistance and improving insulin sensitivity [131]. Therefore, it can help in glucose management in PCOS patients. Moreover, obesity is one of the risk factors for PCOS, and weight management is a crucial component of PCOS treatment [132]. As studies suggest, irisin has a role in reducing body fat and weight loss [133]. Therefore, irisin can play a potential role in the prevention and effective treatment of PCOS. A long-term study on irisin role in PCOS management is also needed. Recent studies have provided significant insights into the role of irisin in managing PCOS. Irisin, a peptide produced during physical exercise, has been linked to various physiological processes that could benefit PCOS patients.

### 4.7. Irisin Role in Follicular Dysfunction Insulin Sensitivity and Metabolic Functions

Exercise-induced irisin has the potential to improve follicular dysfunction in PCOS by inhibiting certain pathways that are involved in the stress response of cells. This can lead to a reduction in inflammation and an improvement in ovarian function [128]. Irisin has been shown to activate the “browning” of adipose tissue, leading to increased heat production and energy expenditure. This could be beneficial in managing insulin resistance and obesity, which are common issues in PCOS. Additionally, irisin′s role in inducing mitochondrial biogenesis can promote better energy utilization in cells, which is crucial for metabolic health in PCOS patients [130]. The ongoing research suggests that irisin could be a promising therapeutic agent for non-communicable diseases, including those related to metabolic syndromes such as PCOS. Its mechanisms involve improving insulin sensitivity, reducing weight, and enhancing cellular metabolism, which are key factors in managing PCOS [127].

These findings underscore the potential of targeting irisin as a part of a therapeutic strategy for PCOS, focusing on its beneficial effects on metabolism, insulin sensitivity, and possibly even reproductive health. Further research is needed to establish more definitive roles and mechanisms by which irisin could be used to treat or manage PCOS effectively.

## 5. Future Study Direction

The irisin role in the HPG axis is still poorly understood, and further studies are needed to find its direct effect on hypothalamus, pituitary, or gonads. Signaling pathways and molecular mechanisms should be further studied to gain insights into irisin’s role in reproductive regulation or their possible use as a therapeutic. Reciprocal hormonal interactions between irisin and other hormones involved in the HPG axis like GnRH, FSH, and LH must be studied to find out the exact mechanism of its role in the HPG axis. Knockout or transgenic models should be used to find out the irisin role in the HPG axis. Understanding its role in detail will help define therapeutic potential of irisin in the HPG axis.

All these studies suggest a significant association of irisin with female reproduction. Its abnormality can be one of the reasons behind female infertility and must be considered while investigating infertility. However, more research is needed to find the exact mechanism. Its receptors are integrin that can alter steroidogenesis in granulosa cells. Therefore, it is suggested that irisin may play a potential role in the pathways of reproductive hormones like estrogen and progesterone. Ovulation, ovarian development, and luteinization, which are all regulated by reproductive hormones, could also be affected because of irisin deletion. The exact mechanism of irisin role in the production of female reproductive hormones must be studied. Therefore, the resulting mechanism of ovarian maturation, ovulation, and menstrual cycle association with irisin can be found. The irisin role in normal reproductive physiology needs to be studied to find the pathophysiology associated with the irisin abnormality. Irisin can be a therapeutic option for an infertile female with reproductive hormones abnormalities.

As the above studies suggest, irisin concentration is significantly associated with fetal growth and birth weight. It is still not clear whether irisin is involved directly or indirectly in fetal birth weight and growth patterns. However, exercise during pregnancy is found to reduce the risk of Macrosomia. Further research is needed to find the efficacy of exercise in preventing growth abnormalities and to establish the type and timing of exercise that can help in the management of fetal birth weight and growth abnormalities. Irisin resistance due to compensatory mechanisms also needs to further investigation, and its association with macrosomia needs to be studied. Therefore, irisin intervention could be considered a possible treatment option for fetal abnormalities related to growth.

Irisin levels during the first trimester can also be used as diagnostic markers for predicting macrosomia or IUGR. However, further research is needed to understand its efficacy and accuracy.

These studies are based on animal trials. Human trials are needed to find the exact optimal beneficial dosage of irisin in reversing the abnormalities and to rule out any side effects on other reproductive organs and hormones that may arise due to irisin treatment. Studies are needed to find the use of irisin in treating endometritis and other inflammatory conditions of the human uterus. Further research is needed to find the mechanism of irisin role in preeclampsia. Irisin levels may be used as a biomarker for preeclampsia and significantly lowered irisin concentration in preeclampsia patients suggests that irisin supplementation can be beneficial in treating or preventing preeclampsia.

Clinical trials are needed to find the therapeutic effects of irisin supplements. Besides that, the optimal, safe, and effective dosage and the timings of irisin supplementation need to be studied. The long-term effects of irisin supplementation on the mother and fetus also need further research.

## 6. Conclusions

Overall, the influence of irisin on female and male reproduction is evident from various studies. However, further research is needed to elucidate irisin mechanism in reproduction and its potential as a therapeutic or diagnostic tool for reproductive dysfunctions and infertility.

## Figures and Tables

**Figure 1 biomolecules-14-01222-f001:**
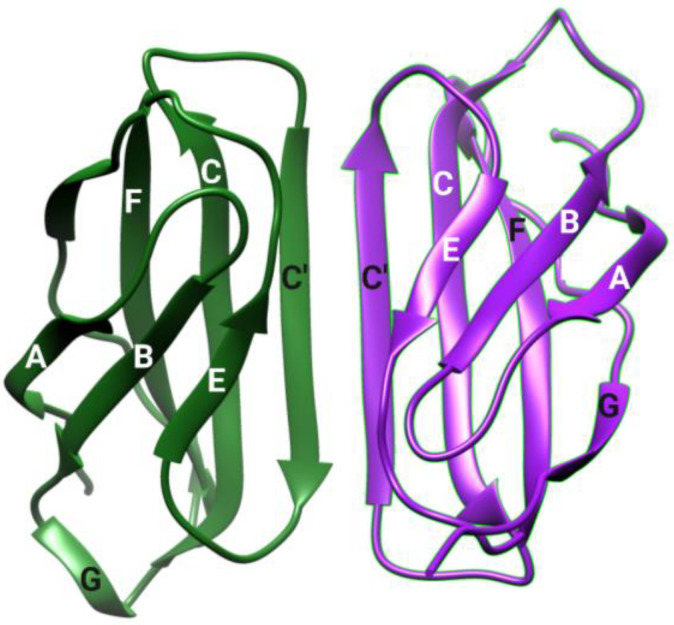
Structure of irisin dimer (*PDB, Chimera*).

**Figure 2 biomolecules-14-01222-f002:**
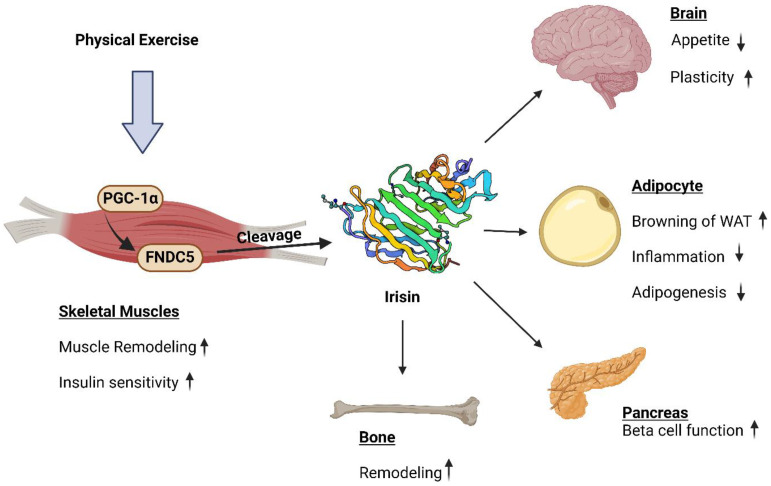
Mechanism and functions of irisin cleaved from FNDC5 during exercise and its effect on different parts of the body like skeletal muscles, brain, adipocytes, pancreas, and bone.

**Figure 3 biomolecules-14-01222-f003:**
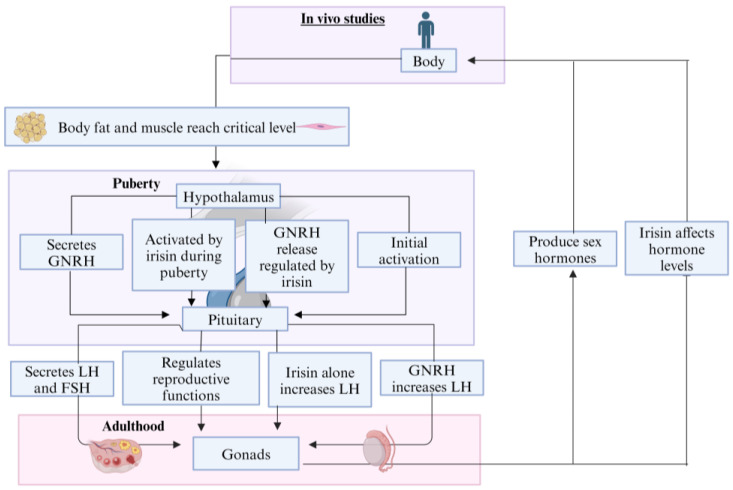
Role of irisin in HPG axis activity.

**Figure 4 biomolecules-14-01222-f004:**
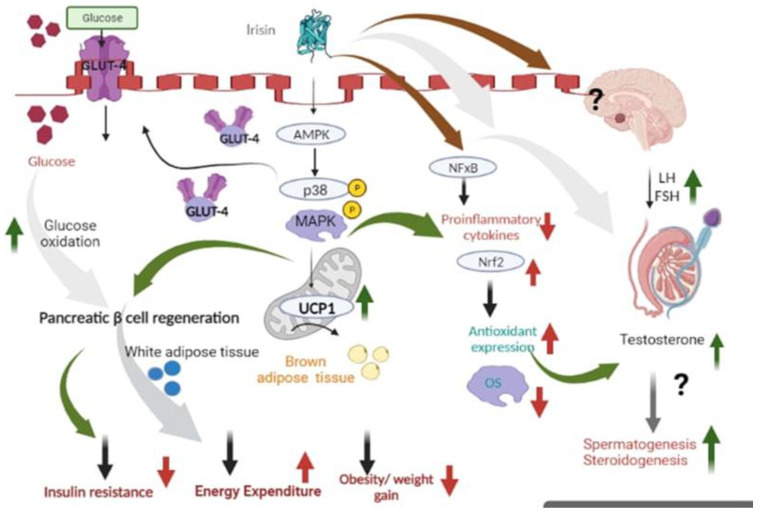
Mechanism of actions of irisin linking energy homeostasis, inflammation, obesity, and male reproduction.

**Figure 5 biomolecules-14-01222-f005:**
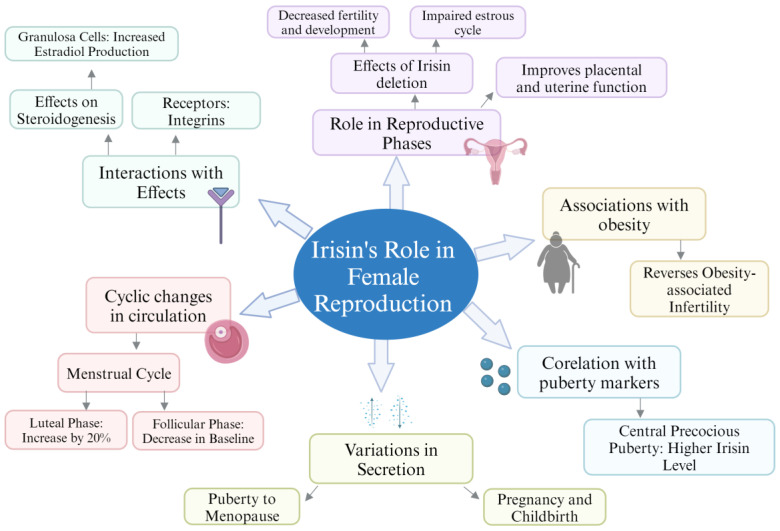
Mind map diagram illustrating irisin role in female reproduction.

**Figure 6 biomolecules-14-01222-f006:**
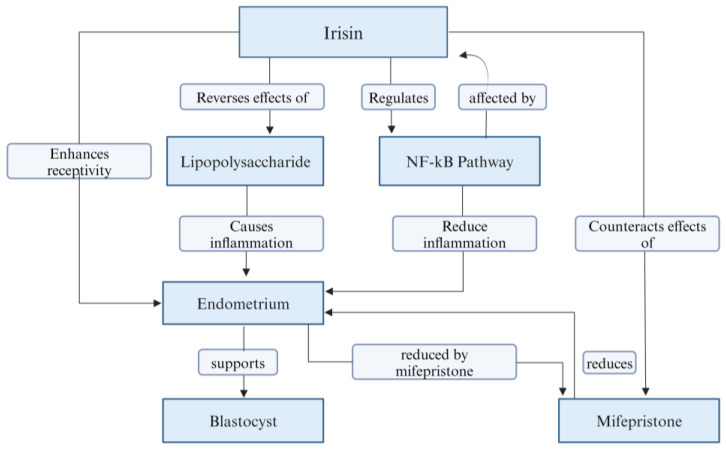
Illustrating the interactions and effects of irisin on the endometrium.

**Figure 7 biomolecules-14-01222-f007:**
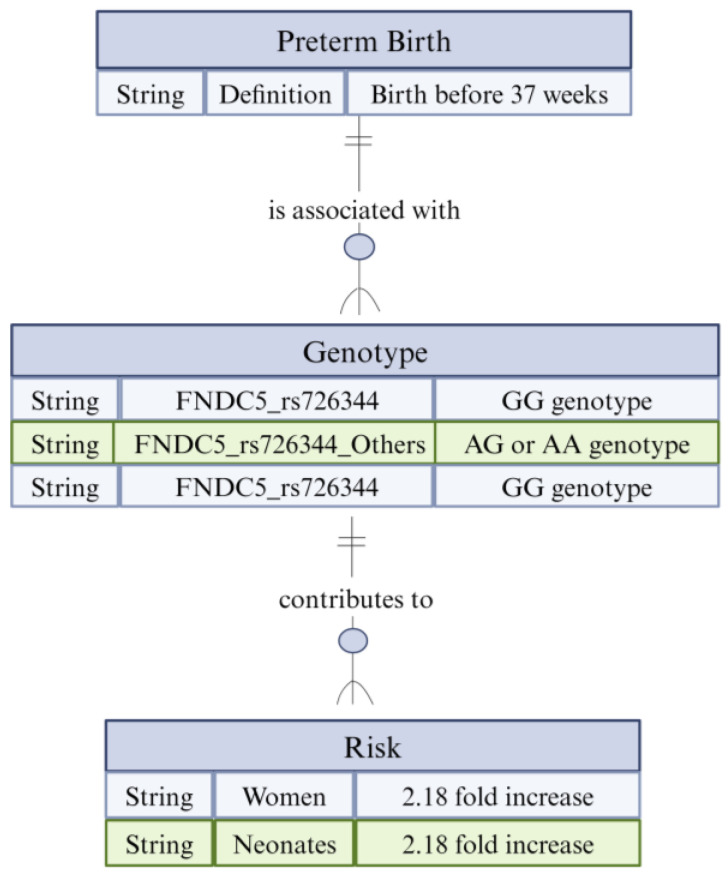
Diagram illustrating the association between the FNDC5 gene polymorphisms and the risk of preterm birth.

**Figure 8 biomolecules-14-01222-f008:**
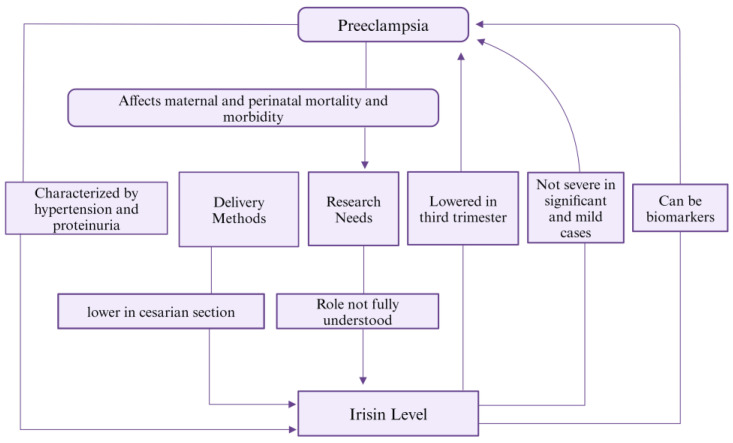
Association of Irisin level and preeclampsia.

**Figure 9 biomolecules-14-01222-f009:**
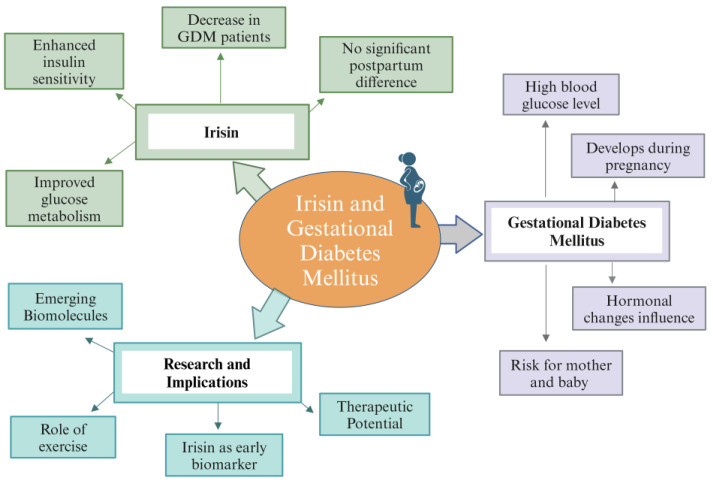
Illustrating the relationship between irisin and gestational diabetes mellitus.

**Table 1 biomolecules-14-01222-t001:** Various functions of irisin in the body.

Function of Irisin	Description	Physiological Impact	Study Reference
Thermogenesis	Irisin is known to convert white fat into brown fat, which increases energy expenditure.	Plays a critical role in regulating body temperature and energy balance.	[33]
Energy homeostasis	Irisin is involved in the regulation of energy intake and expenditure, helping to maintain energy balance.	Crucial for metabolic processes and preventing obesity.	[11]
Browning of white adipose tissue	Irisin stimulates the browning process, which is the conversion of white adipose tissue into metabolically active brown adipose tissue.	May contribute to the reduction in obesity-related disorders.	[34]
Muscle metabolism	Irisin is released during exercise and helps to improve muscle metabolism.	Enhances endurance and strength, contributing to the overall physical fitness.	[35]
Insulin resistance	Irisin has been shown to improve insulin sensitivity, which can help manage blood glucose levels.	Important for the prevention and management of type 2 diabetes.	[36]
Anti-inflammatory effects	Irisin exhibits anti-inflammatory effects in various tissues.	Might be beneficial in combating chronic inflammation-related diseases.	[37]
Neuroprotection	Irisin may play a role in protecting neurons and supporting brain health.	Could be significant for the treatment of neurodegenerative diseases.	[38]
Bone remodeling	Irisin has been implicated in the regulation of bone formation and resorption.	Important for bone health and the potential treatment of osteoporosis.	[39]
Cardiovascular health	Irisin may contribute to cardiovascular health by improving vascular function and reducing arterial stiffness.	Relevant for the prevention of cardiovascular diseases.	[40]

**Table 2 biomolecules-14-01222-t002:** Role of Irisin in female reproduction.

Aspect of Reproduction	Effect of Irisin	Study/Context	References
Menstrual cycle	Irisin levels increase by 25% during the luteal phase compared to the follicular phase. Levels return to baseline during the early follicular phase.	Observational study on menstrual cycle phases	[62]
Ovarian function	Integrins as receptors affect steroidogenesis in ovarian granulosa cells. Increases estradiol production significantly when combined with insulin.	In vitro study on human ovarian granulosa cells	[65,93]
Placental and uterine functions	Improves function in pregnant women.	General observation	[93]
Puberty and pregnancy	Irisin levels increase during puberty and pregnancy, linked to activation of the HPO axis.	Observational study	[62]
Fertility and development	Irisin deletion in mice leads to decreased fertility, impaired ovarian development, and changes in hormone levels.	Experimental study in mice	[4,93]
Obesity and infertility	Irisin treatment reduces serum luteinizing and insulin levels in obese female mice, potentially reversing obesity-associated infertility.	Experimental study in obese mice	[16]
Central precocious puberty	Raised irisin levels are correlated with puberty markers. It suggests a role in the reactivation of the HPG axis and onset of puberty.	Observational study on girls with precocious puberty	[95]

**Table 3 biomolecules-14-01222-t003:** Irisin predictive roles in fetal growth patterns.

Condition	Irisin Role	Observations	References
**Macrosomia**	Higher maternal serum irisin levels in the first trimester are associated with an increased risk of macrosomia.	Positive correlation between higher maternal irisin levels and larger birth weight of the fetus.	[26,100]
**Fetal growth**	Altered irisin levels in maternal and fetal blood correlate with fetal growth patterns.	Lower irisin levels in cases of IUGR; higher levels in macrocosmic infants.	[25,26,101]
**Predictive marker**	Irisin is suggested as a prominent marker for predicting macrosomia over other biomarkers like PIGF.	Irisin′s predictive accuracy for macrosomia is highlighted due to its significant variation in levels.	[26,102]

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
