# Peer review of "Irisin in Reproduction: Its Roles and Therapeutic Potential in Male and Female Fertility Disorders"

_biomolecules, 2024, doi:10.3390/biom14101222_

Round 1

Reviewer 1 Report

Comments and Suggestions for Authors

Comments on the Quality of English Language

The quality of English in this manuscript is adequate, only minor editing.

Author Response

Comment 1: The male section of this review is minimal.

Response: Thank you for your insightful suggestion. We have expanded the male section by including additional studies and details regarding irisin's role in male reproduction, spermatogenesis, and testosterone regulation. Relevant findings have been integrated to provide a more comprehensive overview.

Comment 2: Combine sections on pregnancy, fetal birth weight, endometrium, preterm birth preeclampsia, and GDM.

Response: Thanks for your productive comment. We have combined these topics into two main sections titled "Pregnancy" and "Gestational Disorders" to enhance the manuscript's flow and coherence.

Comment 3: The PCOS section is hard to follow and could be combined into a single section without subsections.

Response: We appreciate your valuable input. We have restructured the PCOS section into a single cohesive section and removed unnecessary subsections to improve readability.

Specific Comments:

  1. Line 7: Comment: Should be "focused" instead of "focus."

Response: Thank you for pointing this out. We have corrected it to "focused."

  1. Line 41: Comment: Should be "direct and indirect" instead of "directly and indirectly."

Response: We appreciate your attention to detail. We have corrected it to "direct and indirect."

  1. Line 104: Comment: There is a repeat of "is a."

Response: Thanks for noticing this. We have removed the repeated phrase.

  1. Line 137: Comment: Missing a "the" after "is."

Response: Thank you for your careful review. We have added "the" to the sentence.

  1. Line 196-197: Comment: There is a floating period after the 71 reference.

Response: We appreciate your keen observation. We have removed the floating period.

  1. Line 324: Comment: Why is LIF defined here since it has already been used multiple times.

Response: Thank you for pointing this out. We have removed the redundant definition.

  1. Line 492-494: Comment: Preeclampsia is hyphenated here but not previously.

Response: We appreciate your attention to consistency. We have standardized the term to "preeclampsia" throughout the manuscript.

  1. Line 499: Comment: Conclusions section should be 6 as future direction is already 5.

Response: Thank you for your suggestion. We have corrected the numbering of the sections.

Reviewer 2 Report

Comments and Suggestions for Authors

Comments to the manuscript “Irisin in Reproduction: Its Roles and Therapeutic Potential in Male and Female Fertility Disorders”.

This is an excellent, comprehensive review about the roles of irisin in reproduction. The manuscript is very well written. The authors have provided evidence from his group of study and others about irisin's role in improving spermatogenesis and testosterone levels, and its correlation with critical reproductive stages (menstrual cycles, puberty, and pregnancy) and conditions like pregnancy complications (preeclampsia, gestational diabetes mellitus) , precocious puberty, PCOS, endometrial receptivity and endometritis. Several high-quality figures have been included in the manuscript, making it easier to read and understand. The authors also highlight the needing of further research to elucidate irisin’s mechanism in reproduction and its potential as a therapeutic or diagnostic tool for reproductive dysfunctions. I recommend acceptation of the manuscript for publication in biomolecules without major changes.

Minor comments

-Introduction, line 35: Please note that definition of FNDC5 is repeated.

-Line 63, please correct “inutero”

-Line 162, please correct “invitro”

-Line 422, please correct “Melitus”

Author Response

Response to Reviewer 2

Minor Comments:

  1. Line 35: Comment: Definition of FNDC5 is repeated.

Response: Thank you for your keen observation. We have removed the repeated definition.

  1. Line 63: Comment: Please correct "inutero."

Response: Thanks for pointing this out. We have corrected it to "in utero."

  1. Line 162: Comment: Please correct "invitro."

Response: We appreciate your attention to detail. We have corrected it to "in vitro."

  1. Line 422: Comment: Please correct "Melitus."

Response: Thank you for your careful review. We have corrected it to "Mellitus."

Reviewer 3 Report

Comments and Suggestions for Authors

This is a very thorough review of irisin and its potential roles in reproductive function. Overall, the manuscript is well organized. Please consider the following to improve readability:

·      The manuscript is a little bit too long, and some of the important points might be lost due to the length of the paper. Please try to condense the manuscript a bit where possible.  

·      Line 11 and line 137: please correct to read: “by influencing the hypothalamic-pituitary-gonadal axis.”

·      Lines 37-39: consider describing the role of irisin (in energy homeostasis, tissue browning, etc.), beyond that it has a role.

·      Line 47: If something is not significant, than we cannot say that “it lowers elevated serum testosterone”’. Please remove this sentence.

·      Line 264: What is HOMA IR? Please describe in the text.

·      Line 284: Please confirm the spelling of “Al Miani”

·      Line 302: Please put a period after (24) .

·      Figure 4.2: If you want to reduce the length of the manuscript, you could remove this figure. Some of the information is covered in figures 2, 3, and table 1, and figure 4.2 doesn’t necessarily add much.

·      Figure 6 and Table 5: these could be combined into one figure or table, as there is redundant information.

Comments on the Quality of English Language

Overall, the English is fine. There are a few minor errors.

Author Response

Response to Reviewer 3

General Comments:

  1. Condense the Manuscript: Comment: The manuscript is a little bit too long.

Response: Thank you for your valuable suggestion. We have condensed the manuscript by removing redundant information and combining overlapping sections and figures.

  1. Line 11 and 137: Comment: Please correct to read "by influencing the hypothalamic-pituitary-gonadal axis."

Response: Thank you for pointing this out. We have corrected it as suggested.

  1. Lines 37-39: Comment: Consider describing the role of irisin beyond its role.

Response: Thanks for your productive comment. We have expanded the description to include irisin's role in energy homeostasis and tissue browning.

  1. Line 47: Comment: Remove the sentence "it lowers elevated serum testosterone."

Response: Thank you for your suggestion. We have removed the sentence as suggested.

  1. Line 264: Comment: What is HOMA IR? Please describe in the text.

Response: Thank you for your insightful comment. We have added a description of HOMA-IR (Homeostatic Model Assessment of Insulin Resistance).

  1. Line 284: Comment: Please confirm the spelling of "Al Miani."

Response: Thank you for your careful review. We have verified and corrected the spelling to "Al-Maini EH."

  1. Line 302: Comment: Please put a period after (24).

Response: Thank you for your observation. We have added the period.

  1. Figure 4.2: Comment: Consider removing this figure.

Response: Thank you for your suggestion. We have removed Figure 4.2 to reduce the length.

  1. Figure 6 and Table 5: Comment: These could be combined into one figure or table.

Response: Thanks for your kind suggestion. We have removed the table and retained only Figure 6 to reduce redundancy.

Reviewer 4 Report

Comments and Suggestions for Authors

This review manuscript describes the myokine irisin and its role in male and female reproduction, along with pregnancy and female fertility disorders. Overall this is a very comprehensive review of previous work related to irisin, with helpful flow diagrams describing proposed mechanisms of action and/or connections between concepts for each section. There are a few minor comments that will need addressed:

Line 32: please rewrite or rearrange the sentence to be more clear to the reader? The protein name (FNDC5) seemed out of place in this sentence.

Line 56: preeclampsia should not be capitalized, and removed the word "pregnancy" after preeclampsia from this sentence.

Line 96: please begin the sentence with "Irisin" instead of "It" to be clear about what is being described/discussed.

Line 103: please define BDNF at the end of the sentence, as this is the first time the acronym is being used.

Line 114: please be consistent with whether or not Irisin is capitalized within a sentence.

Figure 4: please improve the resolution of this figure.

Line 244: soften the language on some of these conclusion sentences. Irisin has not been definitively shown to reverse obesity-associated infertility in all species and conditions (another example is on line 416).

Tables 2-7: these tables are redundant with either the short paragraphs within the manuscript, or the diagrams describing the same concepts. These tables can be removed or moved to supplemental.

Line 260: this section could be combined with the previous section (female reproduction)

Line 285: please clarify what "Al Miana studies" means.

Line 387: the subsections 4.8.1, 4.8.1, and 4.8.3 might not be necessary within the overarching section as the subsections only contain 1-3 sentences each.

Line 485: should the word "define" or "understand" replace "rule out"?

Author Response

Response to Reviewer 4

General Comments:

  1. Line 32: Comment: Rewrite the sentence to be clearer.

Response: Thank you for your suggestion. We have rewritten the sentence for clarity.

  1. Line 56: Comment: Preeclampsia should not be capitalized and remove "pregnancy" after preeclampsia.

Response: Thanks for pointing this out. We have corrected it as suggested.

  1. Line 96: Comment: Begin the sentence with "Irisin" instead of "It."

Response: Thank you for your comment. We have corrected it to start with "Irisin."

  1. Line 103: Comment: Define BDNF at the end of the sentence.

Response: Thanks for your suggestion. We have added the definition of BDNF.

  1. Line 114: Comment: Be consistent with whether or not Irisin is capitalized.

Response: Thank you for your observation. We have standardized the capitalization of "irisin."

  1. Figure 4: Comment: Improve the resolution of this figure.

Response: Thanks for your comment. We have improved the resolution of Figure 4.

  1. Line 244: Comment: Soften the language on some of these conclusion sentences.

Response: Thank you for your suggestion. We have revised the language to be more cautious and precise.

  1. Tables 2-7: Comment: These tables can be removed or moved to supplemental.

Response: Thanks for your comment. We have moved the table 4,5,6,7 to the supplemental section to reduce redundancy.

  1. Line 285: Comment: Please clarify what "Al Miana studies" means.

Response: Thank you for pointing this out. We have clarified the reference to "Al-Maini studies."

  1. Line 387: Comment: Subsections might not be necessary within the overarching section.

Response: Thank you for your suggestion. We have removed unnecessary subsections to streamline the text.

  1. Line 485: Comment: Should the word "define" or "understand" replace "rule out"?

Response: Thanks for your productive comment. We have replaced "rule out" with "understand."